# Harnessing digital health to objectively assess cancer-related fatigue: The impact of fatigue on mobility performance

Yvonne H. Sada[1,2], Olia Poursina[3], He Zhou[3], Biruh T. Workeneh[4], Sandhya V. Maddali[3], Bijan Najafi[3]*

1 Department of Medicine, Section of Hematology and Oncology, Dan L Duncan Cancer Center, Baylor College of Medicine, Houston, Texas, United States of America, 2 Houston VA Center for Innovations in Quality, Effectiveness, and Safety, Michael E. DeBakey VA Medical Center, Houston, Texas, United States of America, 3 Michael E. DeBakey Department of Surgery, Interdisciplinary Consortium on Advanced Motion Performance (iCAMP), Baylor College of Medicine, Houston, Texas, United States of America, 4 Department of Nephrology, Division of Internal Medicine, MD Anderson Cancer Center, Houston, Texas, United States of America

* najafi.bijan@gmail.com

**Data Availability Statement:** All relevant data are within the paper and its Supporting information files. Additional data not directly related to the results of this study (e.g., additional clinical and

## Abstract

### Objective

Cancer-related fatigue (CRF) is highly prevalent among cancer survivors, which may have long-term effects on physical activity and quality of life. CRF is assessed by self-report or clinical observation, which may limit timely diagnosis and management. In this study, we examined the effect of CRF on mobility performance measured by a wearable pendant sensor.

### Methods

This is a secondary analysis of a clinical trial evaluating the benefit of exercise in cancer survivors with chemotherapy-induced peripheral neuropathy (CIPN). CRF status was classified based on a Functional Assessment of Chronic Illness Therapy-Fatigue (FACIT-F) score $\leq$ 33. Among 28 patients (age = 65.7±9.8 years old, BMI = 26.9±4.1kg/m$^2$, sex = 32.9% female) with database variables of interest, twenty-one subjects (75.9%) were classified as non-CRF. Mobility performance, including behavior (sedentary, light, and moderate to vigorous activity (MtV)), postures (sitting, standing, lying, and walking), and locomotion (e.g., steps, postural transitions) were measured using a validated pendant-sensor over 24-hours. Baseline psychosocial, Functional Assessment of Cancer Therapy–General (FACT-G), Falls Efficacy Scale–International (FES-I), and motor-capacity assessments including gait (habitual speed, fast speed, and dual-task speed) and static balance were also performed.

### Results

Both groups had similar baseline clinical and psychosocial characteristics, except for body-mass index (BMI), FACT-G, FACIT-F, and FES-I ($p$<0.050). The groups did not differ on motor-capacity. However, the majority of mobility performance parameters were different

demographic information) are available upon request.

**Funding:** This study was supported by the National Institutes of Health/National Cancer Institute (award number 1R21CA190933-01A1), the National Institutes of Health/National Institute on Aging (award number 1R42AG060853-01), and internal support from Baylor College of Medicine. There was no additional external funding received for this study. The content is solely the responsibility of the authors and does not necessarily represent the official views of sponsors.

**Competing interests:** The authors have declared that no competing interests exist.

between groups with large to very large effect size, Cohen's d ranging from 0.91 to 1.59. Among assessed mobility performance, the largest effect sizes were observed for sedentary-behavior ($d = 1.59$, $p = 0.006$), light-activity ($d = 1.48$, $p = 0.009$), and duration of sitting +lying ($d = 1.46$, $p = 0.016$). The largest correlations between mobility performance and FACIT-F were observed for sitting+lying (rho = -0.67, $p<0.001$) and the number of steps per day (rho = 0.60, $p = 0.001$).

## Conclusion

The results of this study suggest that sensor-based mobility performance monitoring could be considered as a potential digital biomarker for CRF assessment. Future studies warrant evaluating utilization of mobility performance to track changes in CRF over time, response to CRF-related interventions, and earlier detection of CRF.

## Introduction

Cancer-related fatigue (CRF) is defined as unusual tiredness related to cancer or cancer therapy that negatively impacts functional performance [1]. CRF is one of the most common problems reported by cancer patients, and one-third of cancer survivors continue to experience CRF up to 6 years after primary treatment [2]. CRF has been associated with increased psychosocial distress, sedentary behavior, and poor work performance [3, 4]. In addition, older patients with cancer have reported high levels of CRF, which can be complicated by polypharmacy, neuropathy, depression, sleep disorder, frailty, and functional disability [5]. Severe CRF is also an independent predictor of poor survival outcomes [1, 6].

CRF is an important component of cancer-related treatment decision making, such as treatment intensity, dose modifications, referral for psychosocial support, or exercise interventions [7, 8]. However, patient and providers factors may limit accurate CRF assessment. Factors that contribute to variable CRF assessment include lack of time to discuss fatigue during clinic visits, inaccurate reporting because of fear of not receiving maximum cancer treatment and limitations of current CRF measurement tools [9–11]. CRF assessment questionnaires are predominantly subjective and can be biased by the following: 1) patient recall, particularly in older patients with cognitive impairments; 2) inaccurate reporting by patients, who may think their responses may affect treatment decision-making or change how providers perceive them; and 3) inadequate time for clinical practices to perform validated CRF assessment regularly [7].

Unfortunately, most fatigue documentation is based on clinician observation and history-taking skills, which can vary based on provider, rather than validated questionnaires [11]. Even in the clinical trial setting, poor agreement between physician and patient fatigue rating has been reported [11]. Due to the limitations of current CRF assessment tools, improved CRF assessment methods are needed.

Although exercise interventions have been shown to decrease CRF, there is a paucity of data regarding the correlation between mobility performance metrics and CRF [12, 13]. Few studies have attempted to evaluate CRF by assessing motor capacity, such as assessing balance, gait, functional reach, and Timed Up and Go [14]. These assessments are reliable since they are performed in a standardized fashion to control for confounding. However, they do not have high construct validity in CRF [15]. Supervised assessment of motor-capacity may not be

practical for routine screening of CRF because it may overburden the clinical staff and space limitations to administer these tests. In addition, motor-capacity may not accurately represent mobility performance among older and frail adults [16]. The International Classification of Functioning, Disability and Health was introduced by the World Health Organization to differentiate between assessments in a standardized-environment that measure motor capacity, which indicates the highest possible level of function at a specific moment, versus real-life assessments that measure mobility performance, which reflects what individuals do in their natural environment [17].

Digital biomarkers are objective, quantifiable physiological and behavioral data collected and measured by digital devices such as wearable sensors, and applied to the advancement of health [18]. Digital biomarkers of mobility performance may provide a more reliable estimate of physical activity, which is important for clinical decision-making and evaluation of treatment-related adverse effects [19, 20]. The rise of digital health technology, such as fitness and activity trackers, allows monitoring of mobility performance metrics in real-world conditions at a more granular level and can display changes over time. However, the optimal integration of digital biomarkers in clinical decision making remains unknown [18]. There has been an increase in the application of digital biomarkers for cancer care in all phases of cancer survivorship, such as assessment of quality of life, performance status, on-treatment toxicity, and long-term changes in mobility performance after treatment [21–23]. The value of remote monitoring by digital health platform has become even more apparent during the COVID-19 pandemic, given the increased difficulty of evaluating patients virtually by telehealth.

At present, little is known about how digital biomarkers can be applied to CRF assessment. We hypothesize that CRF may impact mobility performance without a noticeable impact on motor capacity. Thus, the primary objective of this study is to examine the association between CRF and mobility performance, as well as motor capacity, measured by wearable sensors. Our second hypothesis is that CRF may be associated with increased sedentary behavior, decreased locomotion, and reduce the duration of standing posture.

## Materials and methods

### Participants

This study is a secondary analysis of a clinical trial that evaluated the benefit of exercise on adult cancer survivors with chemotherapy-induced peripheral neuropathy (CIPN) who had completed chemotherapy treatment (ClinicalTrials.gov Identifier: NCT02773329). The inclusion criteria were as follows: age 55 years or older; ability to provide written informed consent; diagnosis of current or prior malignancy; completion of chemotherapy; neurotoxic chemotherapy or targeted therapy exposure (agents including platinum-based chemotherapy, vinca alkaloids, taxanes, proteasome inhibitors, and interferons); clinically confirmed CIPN, and ability to walk with or without an assistive device for a minimum of fifteen meters. Participants from the clinical trial with complete daily physical activity (DPA) data were utilized for this secondary analysis. The exclusion criteria included: major known joint problems (e.g., back pain, foot problems such as active ulcers and lower extremities amputation, spinal cord injuries); unstable medical condition or medication that may affect mobility (e.g., use of pain suppressant, or chemotherapy ongoing therapy such as radiotherapy, recent stroke), severe cognitive impairment (e.g., dementia, Parkinson's disease), and severe uncorrected visual, hearing, or vestibular impairment. Informed consent was obtained from all participants prior to screening. This study was approved by the Baylor College of Medicine Institutional Review Board.

## Demographic and clinical characteristics

During the screening process, we recorded age, height, weight, and body mass index. We collected self-reported demographics (e.g., ethnicity and race), medical history including duration and type of cancer, history of falls in the past year, comorbidities, cancer diagnosis, self-report number of prescription medicines and over-the-counter medicines taken per day.

Health-related quality of life was assessed using the Functional Assessment of Cancer Therapy–General (FACT-G) survey [24]. Self-reported pain level was extracted from FACT-G. Plantar numbness severity was evaluated by the vibration perception threshold (VPT) as per prior studies and using established thresholds; a VPT value $\geq 25$ volts was classified as severe plantar numbness [25–27]. The Center for Epidemiologic Studies Depression scale (CES-D) short-version scale was used to identify patients with depression based on a cut-off score $\geq 16$ [28]. The Montreal Cognitive Assessment (MoCA) was used to identify subjects with cognitive impairment based on a cut-off score $\leq 25$ [29]. The Fall Efficacy Scale-International (FES-I) questionnaire to determine concern for falls; participants were classified as having high concern for falling if FES-I $\geq 23$ based on previous studies [30–32]. All questionnaires were administered by the assessor.

## CRF evaluation

All subjects completed the Functional Assessment of Chronic Illness Therapy—Fatigue (FACIT-F), which is a validated questionnaire commonly used for the assessment of CRF in clinical trials [33]. A score of less than 34 is used to determine the presence of CRF [34].

## Functional and motor capacity assessments

The Fried criteria were used to determine frailty status (non-frail, pre-frail, and frail) and five physical frailty phenotypes, which includes unintentional weight loss, weakness (grip strength), slow gait speed (15-foot gait test), self-reported exhaustion, and self-reported low physical activity [35]. Subjects with 1 or 2 positive criteria were considered pre-frail, and those with 3 or more positive criteria were deemed to be frail [35]. Subjects who were negative for all criteria were deemed to be robust [35].

Motor capacity was assessed by evaluating standing balance and gait performance based on protocols previously described [25]. Gait and balance were quantified using the LEGSys™ and BalanSens™ (Biosensics LLC, Watertown, MA, USA), respectively. Both platforms use the same hardware configuration of five wearable inertial sensors attached to each subject's shins, thighs, and lower back [36–38]. Gait tests were performed under habitual walking (walking at habitual speed without distraction), dual-task walking (walking + working memory test), and fast walking. Subjects were first asked to walk with habitual gait speed for 15 meters without any distraction (single-task walking). They were asked to walk again while counting backward loudly from a random number (dual-task walking: motor task + working memory). Finally, subjects were asked to walk one more time as fast as they can without running (fast walking). Gait speed was calculated during the steady-state phase of walking using validated algorithms [39, 40]. Standing balance was measured using the same wearable sensors attached to the lower back and dominant front lower shin. Subjects stood in the upright position, keeping feet close together but not touching, with arms folded across the chest for 30 seconds. The center of mass sway (unit: cm$^2$) was calculated using validated algorithms [41].

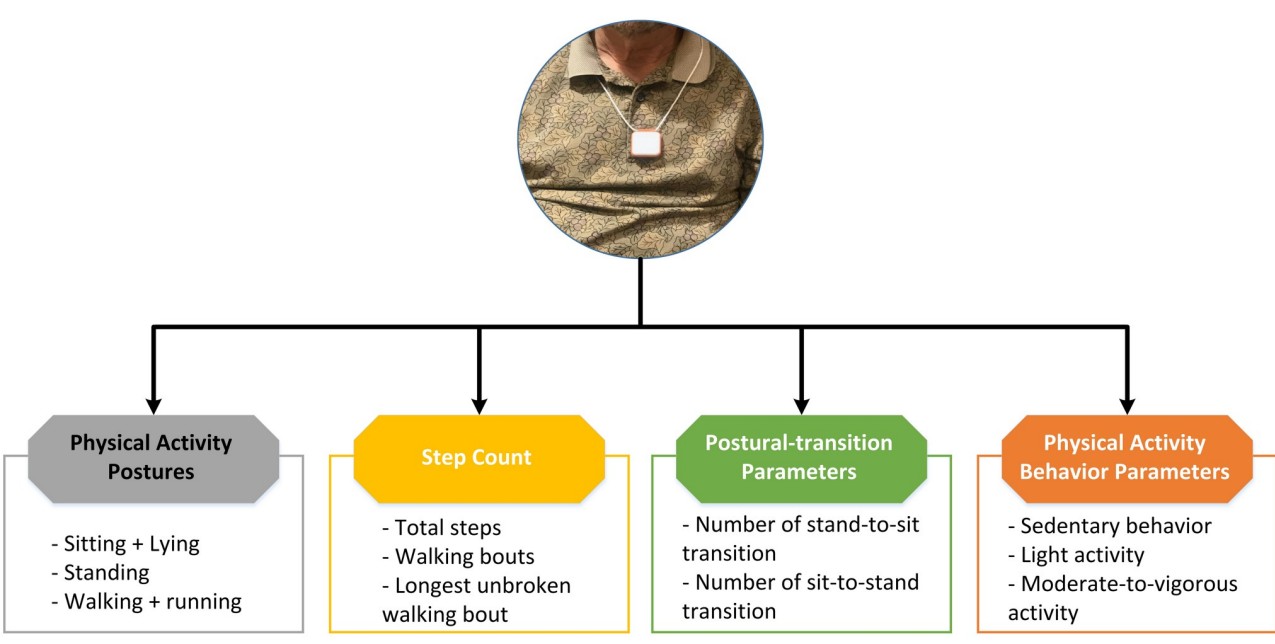

**Fig 1. A wearable pendant sensor was used to monitor Daily Physical Activity (DPA).**

## Assessment of mobility performance using a pendant sensor

Mobility performance was assessed using a pendant sensor described in our previous study [20, 42]. After finishing clinical and functional assessments, the subject was given a wearable sensor (PAMSys™, BioSensics LLC, MA, USA), which can be worn as a pendant (Fig 1), to record daily physical activity (DPA). The subject was instructed to wear the sensor for at least 48-hours continuously (during waking hours and while asleep), then take the sensor off and mail it back to us in a pre-paid envelope. When we received the sensor, we extracted the DPA data stored in the sensor and analyzed the first 24-hours of valid data. The choice of 24-hours was because several subjects did not wear the sensor for 48-hours, but most wore the sensor for the first 24-hours.

The PAMSys™ sensor contains a 3-axis accelerometer (sampling frequency of 50 Hz) and built-in memory for recording long-term data. A previously developed and validated computer program was used to identify body postures, including lying, sitting, standing, and walking [42–45]. The computer program also calculates walking bouts, step counts, and postural transitions, which includes stand-to-sit and sit-to-stand. In this study, we also developed a computer program to calculate sedentary behavior, light activity, and moderate-to-vigorous activity during the daytime (from 10 a.m. to 10 p.m.). High sensitivity, specificity, and accuracy have been reported for the PAMSys™ sensor for the identification of body postures and postural transitions in older adults [42–47]. In this study, daily duration of postures (lying + sitting, standing, and walking + running, as a percentage), activity level as daily percentage, number of walking bouts and steps, stand-to-sit and sit-to-stand postural transitions, and average duration of postural transitions were calculated.

## Statistical analysis

All continuous data were presented as mean ± standard deviation. All categorical data were expressed as a percentage. One-way analysis of covariance (ANCOVA) for normally

distributed variables or Kruskal-Wallis H test for non-normally distributed variables was used to estimate differences of mean between-group comparison of continuous demographic, clinical, and functional performance data. The chi-square test was performed for comparison of categorical demographic, clinical and functional performance data. For between groups comparison for the DPA parameters, we adjusted the results for age, BMI, and FES-I. A 2-sided $p<0.050$ was considered statistically significant. The effect size for discriminating between groups was estimated using Cohen's $d$ effect size and represented as $d$ in the Results section. Values were defined as small (0.20–0.49), medium (0.50–0.79), large (0.80–1.29), and very large (above 1.30) [48]. Values of less than 0.20 were classified as having no noticeable effect [48]. The Spearman rank correlation coefficient was used to evaluate the degree of agreement between the FACIT-F score and daily physical activity parameters. All statistical analyses were performed using IBM SPSS Statistics 25 (IBM, IL, USA).

## Results

We identified 36 adult cancer survivors. Because of technical problems (e.g., battery problem, not recording data, etc) or duration of recording less than 24-hours, the data from three subjects in the non-CRF group and five subjects in the CRF group were excluded. There were 28 patients with complete DPA data (age = 65.7±9.8 years old, BMI = 26.9±4.1kg/m$^2$, sex = 32.1% female). Twenty-one subjects (75.0%) were classified as non-CRF, and the remainder were classified as CRF. Table 1 summarizes demographic and clinical data. The average FACIT-F score of the CRF group was significantly lower than the non-CRF group ($p<0.001$). The CRF group also had a significantly lower FACT-G score than the non-CRF group ($p = 0.027$). Age, ethnicity, and race did not differ significantly between the two groups. There were no women in the CRF group, but the non-CRF group was 42.9% female (p = 0.035).

Table 2 summarizes functional and motor capacity characteristics of study participants. There were no between-group differences regarding history of falling and distribution of frailty status (p>0.050). In addition, aligned with the initial hypothesis, there were no between group differences regarding motor capacity metrics (i.e, habitual walking speed, dual task walking speed, fast walking speed, and static balance).

Table 3 summarizes between group differences for sensor-derived mobility performance metrics. Overall, the CRF group was less active than the non-CRF group. The CRF group spent a significantly higher percentage of time in sitting and lying positions (15.8%, $d = 1.46$, $p = 0.016$), but spent significantly lower percentage of time standing (-49.7%, $d = 1.44$, $p = 0.014$), as well as walking or running (-53.4%, $d = 1.26$, $p = 0.042$). The CRF group had 25.4% more sedentary behavior ($d = 1.59$, $p = 0.006$) but 52.5% less light activity ($d = 1.48$, $p = 0.009$) and 74.6% moderate-to-vigorous activity ($d = 1.40$, $p = 0.020$) when comparing to the non-CRF group. Patients with CRF also had less maximum steps per bout (-56.4%, $d = 1.39$, p = 0.024), and steps (-54.4%, $d = 1.24$, p = 0.047). A non-significant trend towards longer durations of stand-to-sit and sit-to-stand transitions were observed in the CRF group.

Fig 2 demonstrates the correlation between the FACIT-F score and daily PA parameters. In Fig 2A, a significant negative correlation could be observed between the FACIT-F score and sitting and lying postural duration among both CRF and non-CRF subjects ($rho = -0.67$, $p<0.001$). In Fig 2B, a significant correlation could be observed between the FACIT-F score and daily step count among both CRF and non-CRF subjects ($rho = 0.60$, $p = 0.001$).

## Discussion

CRF is associated with adverse long-term effects and poorer survival outcomes for cancer survivors [2–6]. Unfortunately, accurate assessment of CRF is compromised by subjective

**Table 1. Demographics and clinical characteristics of the study participants.**

| | Non-Fatigue (n = 21) | Fatigue (n = 7) | *p-value* |
|---|---|---|---|
| **Demographics** | | | |
| Age, *years* | 66.1 ± 6.9 | 64.7 ± 16.6 | 0.763 |
| Sex (Female), % | 9 (42.9%) | 0 | **0.035** |
| Height, *cm* | 170.2 ± 8.7 | 178.6 ± 6.3 | **0.028** |
| Weight, *kg* | 77.6 ± 14.6 | 88.5 ± 18.0 | 0.121 |
| Body Mass Index, *kg/m²* | 26.6 ± 3.4 | 27.8 ± 6.1 | 0.500 |
| Ethnicity | | | |
| Non-Hispanic, % | 18 (85.7%) | 6 (85.7%) | 0.733 |
| Hispanic, % | 3 (14.3%) | 1 (14.3%) | 0.995 |
| Race | | | |
| White, % | 7 (33.3%) | 3 (42.9%) | 0.649 |
| Black or AA, % | 10 (47.6%) | 3 (42.9%) | 0.827 |
| Other, % | 4 (19.1%) | 1 (14.2%) | 0.776 |
| **Clinical characteristics** | | | |
| FACIT-F, *score* | 43.2 ± 6.3 | 22.4 ± 5.8 | **<0.001** |
| FACT-G, *score* | 90.8 ± 11.2 | 78.6 ± 14.3 | **0.027** |
| Pain level, *score* | 1.00 ± 1.05 | 1.00 ± 1.41 | 0.995 |
| Maximum VPT, *Volt* | 24.0 ± 12.1 | 31.0 ± 16.1 | 0.160 |
| Severe plantar numbness, % | 9 (42.9%) | 4 (57.1%) | 0.512 |
| CESD, *score* | 7.3 ± 9.6 | 8.3 ± 6.8 | 0.801 |
| Depression, % | 2 (9.5%) | 1 (14.3%) | 0.724 |
| MoCA, *score* | 24.1 ± 3.8 | 22.4 ± 4.5 | 0.345 |
| Cognitive Impairment, % | 14 (66.7%) | 5 (71.4%) | 0.815 |
| Time till diagnosis of cancer, *year* | 4.2 ± 3.1 | 8.2 ± 5.6 | **0.023** |
| Number of medication per day | | | |
| Prescription medications, *n* | 4 ± 3 | 6 ± 4 | 0.147 |
| Over-the-counter medications, *n* | 1 ± 1 | 1 ± 1 | 0.557 |
| **Comorbidities** | | | |
| High blood pressure, % | 12 (57.1%) | 2 (28.6%) | 0.190 |
| Heart /circulation problem, % | 2 (9.5%) | 0 | 0.397 |
| Musculoskeletal problem, % | 5 (23.8%) | 1 (14.3%) | 0.595 |
| Stroke, % | 1 (4.8%) | 1 (14.3%) | 0.397 |
| Parkinson, % | 0 | 0 | 0.995 |
| Sleep problem, % | 6 (28.6%) | 6 (28.6%) | 0.995 |
| Rheumatoid Arthritis, % | 2 (9.5%) | 6 (28.6%) | 0.212 |
| Diabetes, % | 5 (23.8%) | 6 (28.6%) | 0.801 |

Note: Values are mean ± SD

*n*: number

*s.d.*: standard deviation.

FACIT-F: Functional Assessment of Chronic Illness Therapy–Fatigue, Score of 33 or lower is fatigue.

FACT-G: Functional assessment of Cancer Therapy–General.

Pain level was assessed using FACT-G, Scale 0 (not at all) to scale 4 (very much pain).

VPT: Vibration Perception Threshold, Score of 25 or greater is sever plantar neuropathy.

CESD: Center for Epidemiologic Studies Depression, Score of 16 or greater is clinical depression.

MoCA: Montreal Cognitive Assessment, score of 25 or lower is cognitive impairment.

Significant difference (p<0.050) between groups were indicated in **bold**.

**Table 2. Functional characteristics of the study participants.**

|  | Non-Fatigue | Fatigue | *p-value* |
|---|:---:|:---:|:---:|
|  | **(n = 21)** | **(n = 7)** |  |
| Frailty |  |  |  |
| Robust, % | 11 (53.8%) | 2 (33.3%) | 0.522 |
| Pre-frail, % | 10 (46.2%) | 2 (33.3%) | 0.687 |
| Frail, % | 0 | 2 (33.3%) | **0.032** |
| Fall in the last 12 months |  |  |  |
| 0, % | 16 (76.2%) | 4 (57.1%) | 0.334 |
| 1–3, % | 4 (19.0%) | 2 (28.6%) | 0.595 |
| > 3, % | 1 (4.8%) | 1 (14.3%) | 0.397 |
| FES-I, *score* | 23.5 ± 6.9 | 29.3 ± 7.7 | 0.072 |
| High concern about falling, % | 10 (47.6%) | 5 (71.4%) | 0.274 |
| Habitual walking speed, *m/s* | 1.03 ± 0.21 | 0.84 ± 0.25 | 0.067 |
| Dual task waking speed, *m/s* | 0.91 ± 0.29 | 0.71 ± 0.20 | 0.124 |
| Fast waking speed, *m/s* | 1.37± 0.34 | 1.19 ± 0.33 | 0.254 |
| Static balance (CoM sway), *cm²* | 0.19 ± 0.18 | 0.18 ± 0.12 | 0.830 |

Note: Values are mean ± SD

*s.d.*: standard deviation

FES-I: Falls Efficacy Scale–International, Score of 23 or greater is high concern of falling

CoM: center of mass

m/s:meter/second

*cm²*:square centimeter.

Significant difference (p<0.050) between groups were indicated in **bold**.

screening tools, patient reporting bias, and variable evaluation by clinicians regularly [7, 9–11]. Thus, alternative strategies utilizing digital biomarkers to enhance CRF assessment are being evaluated. The findings from our cross-sectional study of cancer survivors with chemotherapy-induced peripheral neuropathy may inform further development of digital biomarkers as an indicator of CRF. Aligned with primary hypothesis of this study, CRF was not associated with mobility capacity metrics (gait speed and static balance). Whereas it was associated with deterioration in mobility performance including increase in sedentary activities (high sedentary behavior and lower light and MtV activities), increase in cumulative sedentary postures (longer sitting and lying postures and shorter standing posture), and decrease in locomotion activities (lower step count and shorter longest unbroken walking bout).

## Fatigue and sedentary behavior

Our data suggest that CRF is associated with increased sedentary behavior, higher cumulative sedentary postures, lower step counts, and shorter walking bouts. Based on the correlation of step count and shorter walking bouts with FACIT-F, these digital biomarkers could be utilized as an objective digital biomarker that is potentially more sensitive and less biased than self-reported questionnaires [22]. A recent meta-analysis demonstrated that sedentary behavior is associated with increased all-cause and cancer-specific mortality, but inconsistent results regarding fatigue [49]. Previous cross-sectional studies that showed no correlation between fatigue and sedentary behavior are often limited by the use of self-report questionnaires [50]. Few negative studies used objective measurements to quantify physical activities and examined association between sedentary behavior and psychosocial metrics including depression, quality

**Table 3. Between-group comparison for daily physical activity.**

|  | Non-Fatigue (n = 21) | Fatigue (n = 7) | Mean Difference % | *Cohen's d* | *p-value** |
|---|---|---|---|---|---|
| Sitting + lying percentage, % | 76.3 ± 9.6 | 88.3 ± 6.6 | 15.8% | 1.46 | **0.016** |
| Standing percentage, % | 17.1 ± 6.4 | 8.6 ± 5.2 | -49.7% | 1.44 | **0.014** |
| Walking + running percentage, % | 6.6 ± 3.6 | 3.1 ± 1.6 | -53.4% | 1.26 | **0.042** |
| Sedentary behavior, % | 68.7 ± 13.6 | 86.1 ± 7.4 | 25.4% | 1.59 | **0.006** |
| Light activity, % | 26.8 ± 11.4 | 12.7 ± 7.1 | -52.5% | 1.48 | **0.009** |
| Moderate-to-vigorous activity, % | 4.5 ± 3.2 | 1.2 ± 1.3 | -74.6% | 1.40 | **0.020** |
| Walking bouts, *n* | 255 ± 153 | 121 ± 74 | -52.7% | 1.12 | 0.075 |
| Step count, *n* | 5375 ± 3073 | 2450 ± 1319 | -54.4% | 1.24 | **0.047** |
| Maximum number of steps per bout, *n* | 374 ± 200 | 163 ± 77 | -56.4% | 1.39 | **0.024** |
| Number of stand-to-sit transition, *n* | 90 ± 50 | 56± 50 | -38.1% | 0.69 | 0.188 |
| Average duration of stand-to-sit transition, *second* | 2.79 ± 0.14 | 2.96 ± 0.43 | 6.0% | 0.52 | 0.168 |
| Number of sit-to-stand transition, *n* | 101 ± 51 | 55 ± 50 | -45.6% | 0.91 | 0.092 |
| Average duration of sit-to-stand transition, *second* | 2.81 ± 0.18 | 2.97 ± 0.13 | 5.5% | 0.99 | 0.263 |

Note: Values are mean ± SD

*n*: number

*s.d.*: standard deviation.

*: Results were adjusted by age, BMI, and FES-I.

Effect sizes were calculated as *Cohen's d*.

Significant difference (p<0.050) between groups were indicated in **bold**.

of life, and anxiety in cancer survivors [51–53]. However, these studies did not use a validated instrument to quantify CRF, and fatigue status was determined using an indirect method such as a sub-component of the quality of life questionnaire. In contrast, a 7 days monitoring of breast cancer survivors, who used an accelerometer to assess sedentary behavior, demonstrated

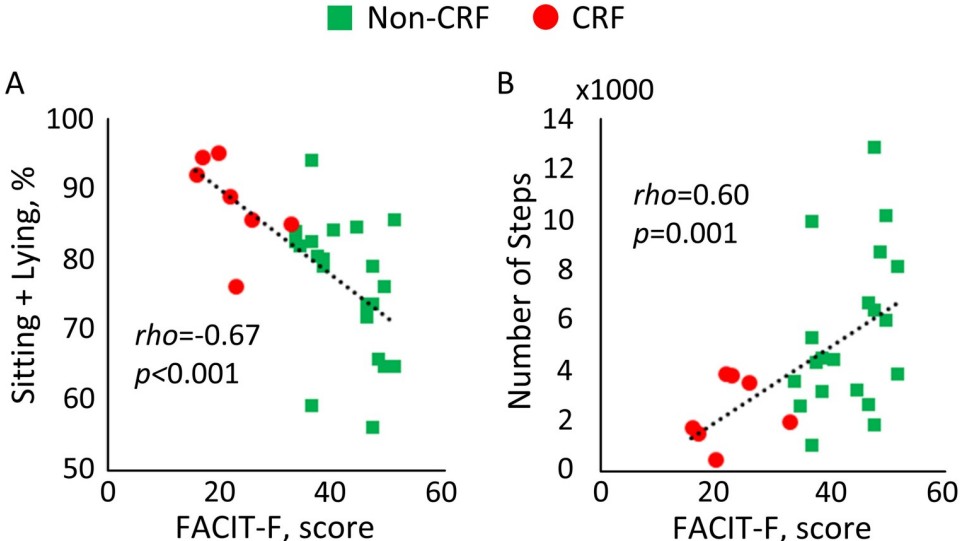

**Fig 2. Agreement between cancer-related fatigue (CRF) and A) sitting+lying postures as a percentage of 24-hour daily physical activity (DPA); and B) number of daily steps per day.**

a correlation of activity with increased fatigue [54]. Similarly, cross-sectional studies of lung cancer and colon cancer survivors have demonstrated a correlation between increased sedentary behavior and fatigue [55–57]. However, these studies lack granularity about activity pattern, such as cumulative postures, postural transitions, and walking bouts.

Walking bout and step count reflect a patient's mobility performance, rather than motor capacity [16]. Interestingly, metrics associated with motor capacity such as walking speed assessments or balance were not associated with CRF suggesting a single time point assessment under supervised conditions may not be sufficient to determine CRF. Prior studies support this hypothesis. A pilot study of patients receiving chemotherapy demonstrated a correlation between step count per day and fatigue [58]. Previous studies of older patients in a community setting have shown that motor capacity assessment does not necessarily predict mobility performance [15, 16, 59, 60].

## Fatigue and posture

Postural transitions and duration of a given posture (lying, sitting, or standing) may also be useful digital biomarkers of mobility performance and indicators of fatigue. To our knowledge, this is the first study that examined the association between CRF and posture as well as postural transition. Prior studies, in which the association between fatigue and physical activity were examined [51, 53], were often focused on sedentary behavior, which was estimated by quantifying acceleration intensity level, called activity count. While there is an overlapping between sedentary behavior and some of the cumulative posture durations (e.g., sitting and lying), some postures such as standing or low speed walking may be categorized as sedentary behavior using activity count. However, it is not necessarily an indicator of poor mobility performance and thus may provide independent information than sedentary behavior [61]. In addition, sedentary behavior represents stationary activities with a duration of over 30 seconds therefore neglect some transition events like postural transitions (e.g., sit-to-stand transitions) [43, 45]. Thus, postural and postural transition data could be a more accurate indicator of mobility performance than sedentary behavior. Our data show that patients with CRF have nearly 38.1% fewer sit-to-stand transitions than patients without CRF. A potential explanation why people with CRF have fewer sit to stand transitions is the need for energy conservation, because patients require more energy expenditure to move from sit-to-standing posture [62]. Studies have shown that cancer survivors do more passive leisure pursuits [63]. Postural transitions rather than total sedentary time alone warrant further study as a dimension of a digital biomarker for CRF. Postural transitions may also serve as an endpoint in CRF related interventions, as reducing sedentary time may be a target to improve quality of life in cancer survivors [55–57, 64, 65].

Our study had several limitations inherent to a cross-sectional study, which cannot evaluate causality. Because this study is a secondary analysis, the study population was limited to cancer survivors with CIPN, and the study was not powered to clinically validate association between CRF and mobility performance metrics. The duration of physical activity monitoring was limited to 24-hours, which may not be sufficient for a reliable assessment of mobility performance. In addition, this study linked the physical activities to self-report fatigue over the week prior, which may not represent fatigue during the period of physical activity monitoring as fatigue levels can change on a daily basis. Patients were evaluated at a single time point that occurred during an extended time period after chemotherapy completion. Although the effect of chemotherapy on CRF during active treatment was not assessed, CRF is a known long-term sequela of chemotherapy that remains an important cancer survivorship issue. The sample size

was small; thus, our findings are hypothesis-generating and require prospective evaluation in a larger cohort.

In conclusion, step count, sedentary cumulative postures (sitting and lying), sedentary behavior, and longest walking bout correlate with CRF in patients with cancer, and may serve as a potential digital biomarkers surrogate of CRF. Our data suggest that digital activity monitors could be considered as a tool to enhance current methods of CRF evaluation. Given the prevalence of CRF among cancer survivors and negative impact on the quality of life, larger cohort studies are needed to validate digital biomarkers of CRF, as well as to integrate digital measures of performance mobility into clinical practice and CRF interventions. In addition, future study is warranted to evaluate the sensitivity to change of these digital biomarkers and their ability to track dynamic changes in CRF over time.

## Supporting information

**S1 Data.**
(XLSX)

## Acknowledgments

We thank Linda Garland, Ana Enriquez, Ivan Marin, Louie Mersey, and Manual Gardea for assisting with data collection, data analysis, and coordination of this research study between involved key investigators.

## Author Contributions

**Conceptualization:** Yvonne H. Sada, Biruh T. Workeneh, Bijan Najafi.

**Data curation:** Yvonne H. Sada, Olia Poursina, Biruh T. Workeneh, Sandhya V. Maddali.

**Formal analysis:** Olia Poursina, He Zhou.

**Funding acquisition:** Bijan Najafi.

**Methodology:** Bijan Najafi.

**Supervision:** Yvonne H. Sada, Bijan Najafi.

**Writing – original draft:** Yvonne H. Sada, Olia Poursina, He Zhou, Biruh T. Workeneh, Sandhya V. Maddali, Bijan Najafi.

**Writing – review & editing:** Yvonne H. Sada, Olia Poursina, He Zhou, Biruh T. Workeneh, Sandhya V. Maddali, Bijan Najafi.

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
