## [Decision Letter · Decision Letter 0]

22 May 2020

PONE-D-20-01496

Harnessing Digital Health to Objectively Assess Cancer-Related Fatigue: The Impact of Fatigue on Mobility Performance

PLOS ONE

Dear Dr. Najafi,

Thank you for submitting your manuscript to PLOS ONE. After careful consideration, we feel that it has merit but does not fully meet PLOS ONE’s publication criteria as it currently stands. Therefore, we invite you to submit a revised version of the manuscript that addresses the points raised during the review process.

Your manuscript has been assessed by four reviewers, who are positive about the manuscript overall. Reviewer 2 has expertise in biostatistics, whereas the other three reviewers are subject experts. The reviewers raise a number of concerns that should be addressed with appropriate revisions, including restricting the demographic data and inter-group difference assessments to just those participants with DPA data, including a sample size and power calculation, improving aspects of the methodology and reporting, and providing further justification for this study.

We look forward to receiving your revised manuscript.

Kind regards,

Emily Chenette

Staff Editor

PLOS ONE

Journal Requirements:

2. Thank you for including your ethics statement:  "Informed consent approved by local Institutional Review Boards was obtained from all participants prior to screening."

"Partial support was provided by the National Institutes of Health/National Cancer Institute (award number 1R21CA190933-01A1). "

Reviewers' comments:

Reviewer's Responses to Questions

**Comments to the Author**

1. Is the manuscript technically sound, and do the data support the conclusions?

Reviewer #1: Partly

Reviewer #2: No

Reviewer #3: Partly

Reviewer #4: Partly

2. Has the statistical analysis been performed appropriately and rigorously? 

Reviewer #1: No

Reviewer #2: No

Reviewer #3: Yes

Reviewer #4: Yes

3. Have the authors made all data underlying the findings in their manuscript fully available?

Reviewer #1: No

Reviewer #2: Yes

Reviewer #3: Yes

Reviewer #4: Yes

4. Is the manuscript presented in an intelligible fashion and written in standard English?

Reviewer #1: Yes

Reviewer #2: Yes

Reviewer #3: Yes

Reviewer #4: Yes

5. Review Comments to the Author

Reviewer #1: Interesting paper on an important topic but limited by a flawed statistical analysis which negatively impacts the integrity of the data and thus the conclusions that can be made. See below for specific comments:

Abstract

The first sentence of the objective implies that CRF is a comorbidity of CIPN, which I don't think is the intention. Please modify this first sentence and section to reflect the primary aims of the study to evaluate mobility in CRF (regardless of comorbidity) and clarify in the methods that this is a secondary analysis of a clinical trial of exercise in CIPN. Additionally, a minor point, but please report 'sex' (physical/biological construct) rather than 'gender' (social construct) unless you are referring to the identity of individuals.

Introduction

2nd paragraph, 1st sentence - please expand on the impact of CRF on treatment decisions (i.e. dose delays/modifications during treatment? treatment decisions only during survivorship?) to illustrate the complete impact of CRF.

3rd paragraph, last sentence - please change to 'Due to the limitations of current CRF assessments tools, improved assessment methods are needed'. No data exist to suggest that objective/quantifiable measures will be inherently better/more valid/more sensitive/more reliable in assessing CRF.

4th and 5th paragraphs - I'd reorganize these paragraphs for clarity in the following manner (essentially flip paragraphs 4 and 5): 1) links between physical activity and CRF and rationale for evaluating CRF; 2) prospective utility of digital biomarkers in evaluating mobility. At present, the utility of biomarkers in evaluating physical activity is presented before a clear rationale for evaluating physical activity in relation to CRF has been established.

5th paragraph, final sentence - This is a cross-sectional study, and your hypothesis should reflect this. '...CRF may >>BE ASSOCIATED WITH<< increased sedentary behavior, decreased locomotion...' Please refrain from implying causation.

Methods

As a general point, the wide range of available assessments is a strength of this cross sectional assessment, as it allows for analysis of the wide range of counfounders. However, assessments that are not relevant to the primary aims can be described more concisely (i.e. 'Plantar numbness severity was evaluated by VPT as per prior studies [26, 27] and using established thresholds [25, 26]).

Statistical Analysis - the ANCOVA should also account for the significant between-group differences in FES-I scores, as an increased fear of falling is associated with decreases in mobility

Results

This section reveals the critical flaw in this study (albeit one that is easily addressable) in paragraph 3 - demographics/clinical characteristics are presented and inter-group differences are calculated for 36 subjects, while DPA data are only available for 28 subjects. Given that the aims and conclusions are all based on DPA data, demographic data and inter-group differences MUST be presented and calculated based on the group of subjects with available DPA data to allow for sufficient analysis of potential confounding factors. This manuscript currently presents demographic/clinical characteristic data (to determine confounders) and DPA data as though they are from the same cohort when they ARE NOT! The authors must address this critical flaw to facilitate valid analysis of data. Several specific comments below:

Paragraph 1, sentence 5 - a cross sectional analysis of 36/28 people cannot be used to suggest that CRF is independent of age, ethnicity and race. Please modify along the lines of ' age, ethnicity and race did not significantly differ between groups.'

Paragraph 1, sentence 7 & 8 - please refrain from reporting non-significant results. The data show that no inter-group differences in self-reported pain levels, plantar numbness, depression, and prescription medications exist.

Paragraph 2, sentence 3 - again, these results are all non-significant. There are no differences in gait speed and static balance between groups. Please delete or revise this sentence.

Table 2 - the analysis of between-group differences in frailty in inappropriate. The point of interest here is a difference in the *distribution* of frailty classifications between groups, not individual comparisons of percentages of robust/pre-frail/frail subjects. Accordingly, a chi-square test should be use to test for a difference in this distribution. It appears that these distributions are different (CRF group more frail), which, if validated by appropriate statistics, is a massive confounder for reported results.

Discussion

This section is liable to change quite significantly when appropriate analysis methods are used, so have must made a few comments. In general, a revised discussion should include a greatly expanded impact of the impact of confounders (as calculated through revised statistical analyses).

Paragraph 2, sentence 1 - again, please refrain from implying causation. CRF is associated with increased sedentary behavior, lower step counts, and shorter walking bouts.

Paragraph 2, sentence 4 - the authors mention the assessment of physical activity levels using accelerometers as a limitation to previous studies, but also a strength of a different prior study in sentence 5. please revise.

Reviewer #2: The manuscript entitled ‘Harnessing Digital Health to Objectively Assess Cancer-Related Fatigue: The Impact of Fatigue on Mobility Performance' with the aim to examine the effect of CRF on mobility performance measured by a wearable pendant sensor which ultimate goal to develop a digital health platform to assess CRF objectively.

The manuscript requires major improvements especially with regards to the data/results presentation. Certain parts of the text in the manuscript need to be written more systematically.

Comments

Materials and Methods

There was no sample size calculation for the study or power of study from the sample size was discussed.

The mode of administration for all the questionnaire/inventories to be clearly stated. i.e. self-administered or by assessor/interviewer.

Statistical analysis

Analysis of Chi-square to be written as chi-square test. If it was referred to the chi-square function presented in the analysis output of a particular statistical test, the statistical test to be stated. Spearman coefficient to be written as Spearman rank correlation coefficient.

Results

Page 11, for the ‘Results suggest that CRF is independent of age, ethnicity and race (p>0.050), more detail results to be provided in a table form.

Page 11, for the ethnicity and race, although they have different meaning, can these two be combined and placed as one variable?

Page 11, for the higher number of prescription medications (50%, p=0.052), the data to be presented in Table 1.

Page 11, for the section ‘Results suggest that CRF is independent of age, ethnicity and race (p>0.050). The CRF group also showed significantly higher BMI (p=0.017) and lower FACT-G score (p=0.004) than the non-CRF group. Average self-reported pain level (assessed using a subcomponent of FACIT-G) had a trend towards higher levels in the CRF group compared to non-CRF (p=0.060). Participants in the CRF group had a non-significant trend towards higher plantar numbness quantified by VPT on average by 29% (p=0.160)’, the figures to be written other than the p value or the figures to be omitted if the figures were listed in the table (for standardization purposes).

Page 11, the statement ‘Participants in the CRF group had a non-significant trend towards higher plantar numbness quantified by VPT on average by 29%’ not clear and figure(s) to be presented in Table 1.

For Table 1, 2, 3, mean, sd and statistical test(s) to be denoted in the table footnote.

For Table 1, 2, n to be stated apart from percentages. The analysis involving categorical variables need to be re-looked. The use of chi square test not clear. Chi-square test is to be employed for categorical data rather than using proportions testing on the variable individual level.

For Figure 2, r symbol was used. Was Pearson's correlation coefficient used in the analysis? If it was meant for Spearman's correlation, the symbol to be replaced with symbol rho.

The percentage figures in the text and tables (Table 1, 2, 3) to be at least one decimal point.

Discussion

Page 17, it was stated 'The sample size was small'. Based on what evidence/ground, this statement was derived?

Page 17 and 18, for the statement 'although we adjusted for several factors such as age, pain level, comorbidities, and depression, we were unable to account for confounding factors such as cancer type, chemotherapy regimen, or dosing. In order to address this limitation, we created three separate regression models.' were these for this study or other study?

References did not conform with the journal format.

Reviewer #3: As mentioned in the paper, cancer-related fatigue (CRF) is a distressing symptom among cancer patients, and there is limited data on effective treatment options. This study examines the effect of CRF on mobility performance measured by a wearable pendant sensor. However, there are some questions/ concerns as follows.

(1) As we know, CRF includes physical fatigue, affective fatigue and cognitive fatigue. Not clear which types of CRF can the wearable pendant sensor measure?

(2)Please describe the measuring timepoint of FACIT-G, only at baseline?

(3)Which is the purpose of this study? the effect of CRF on mobility perfomance? or the accurancy of the wearable pendant sensor? Please clear the purpose which influence the study design directly.

(4) Why “the participants age 55 years or older”? any basis?

(5) “Thirty-six adult cancer survivors with CIPN (age=65.7±9.4 years old,

BMI=27.6±4.4kg/m2, gender=36% female), who completed chemotherapy treatment, were recruited.” should be put in the result parts.

(6) “several subjects did not wear the sensor for 48-hours”, why? If they won’t wear the pendant over 48 hours, how can we use this device to assess the CRF in the daily care of cancer patients?

(7)Limitations may expand after the above comments are addressed.

Reviewer #4: Thanks for the opportunity to review.

I think the paper is presented in a simplistic manner regarding CRF. It is complex and subjective - similar to pain therefore the justification for this research is lacking. Why are objective measures needed? There are ways of measuring effectiveness of interventions. Reasons used for the little attention CRF receives from clinicians is not shared in other countries where cancer rehabilitation is making major gains.

Some other recommended changes are:

Avoid using words such as ‘suffer’ – suggest ‘experience’ or ‘report’.

Why does CRF need to be assessed objectively? Treatment can be evaluated on patient self-report similar to pain. Will objective measures change things for the person with CRF – no.

Inclusion criteria – why did the participants need to be 55 years or older?

Why this limited population – only those with CIPN? Not noted in the limitations

Other differences – number of females – 42% in the non-CRF compared with 25% in the CRF. Frailty differences between groups. Fear of falling – concern about falling has a major impact on mobility.

This is not very person-centred in its approach to CRF.

The reason why people with CRF have fewer sit to stand transitions is about energy conservation. It takes more energy to move from sit to standing. People with CRF change the activities they do in response to CRF. Studies have shown that cancer survivors do more passive leisure pursuits.

Recommendations for future research missing. Power calculations for an adequately sample size?

Correct referencing is needed eg. WHO ICF in the reference list.

6. PLOS authors have the option to publish the peer review history of their article (what does this mean?). If published, this will include your full peer review and any attached files.

Reviewer #1: No

Reviewer #2: No

Reviewer #3: No

Reviewer #4: Yes: Carol McKinstry

---

## [Author Response · Author response to Decision Letter 0]

18 Aug 2020

Please see attached our response letter

---

## [Decision Letter · Decision Letter 1]

14 Jan 2021

Harnessing Digital Health to Objectively Assess Cancer-Related Fatigue: The Impact of Fatigue on Mobility Performance

PONE-D-20-01496R1

Dear Dr. Najafi,

We’re pleased to inform you that your manuscript has been judged scientifically suitable for publication and will be formally accepted for publication once it meets all outstanding technical requirements.

Kind regards,

George Vousden

Senior Editor

PLOS ONE

Additional Editor Comments (optional):

Please address the minor comments below.

Reviewers' comments:

Reviewer's Responses to Questions

**Comments to the Author**

1. If the authors have adequately addressed your comments raised in a previous round of review and you feel that this manuscript is now acceptable for publication, you may indicate that here to bypass the “Comments to the Author” section, enter your conflict of interest statement in the “Confidential to Editor” section, and submit your "Accept" recommendation.

Reviewer #1: All comments have been addressed

Reviewer #2: (No Response)

2. Is the manuscript technically sound, and do the data support the conclusions?

Reviewer #1: Yes

Reviewer #2: No

3. Has the statistical analysis been performed appropriately and rigorously? 

Reviewer #1: Yes

Reviewer #2: No

4. Have the authors made all data underlying the findings in their manuscript fully available?

Reviewer #1: Yes

Reviewer #2: Yes

5. Is the manuscript presented in an intelligible fashion and written in standard English?

Reviewer #1: Yes

Reviewer #2: Yes

6. Review Comments to the Author

Reviewer #1: An excellent and comprehensive revision resulting in a much stronger manuscript - no further comments from me

Reviewer #2: Minor comments

Line 200-207, for the benefit of readers, the statistical tests name mentioned here to be clearly denoted in the tables footnote.

Line 218, word mean, sd to be stated.

7. PLOS authors have the option to publish the peer review history of their article (what does this mean?). If published, this will include your full peer review and any attached files.

Reviewer #1: **Yes: **J. Matt McCrary

Reviewer #2: No

---

## [Editor Report · Acceptance letter]

19 Feb 2021

PONE-D-20-01496R1 

Harnessing Digital Health to Objectively Assess Cancer-Related Fatigue: The Impact of Fatigue On Mobility Performance 

Dear Dr. Najafi:

I'm pleased to inform you that your manuscript has been deemed suitable for publication in PLOS ONE. Congratulations! Your manuscript is now with our production department. 

Kind regards, 

on behalf of

Dr. George Vousden 

Staff Editor

PLOS ONE